# *Candida albicans* as a Trailblazer for Herpes Simplex Virus-2 Infection Against an In Vitro Reconstituted Human Vaginal Epithelium

**DOI:** 10.3390/microorganisms13040905

**Published:** 2025-04-14

**Authors:** Francesco Ricchi, Stefania Caramaschi, Arianna Sala, Laura Franceschini, Luca Fabbiani, Andrea Ardizzoni, Elisabetta Blasi, Claudio Cermelli

**Affiliations:** 1Clinical and Experimental Medicine PhD Program, Department of Biomedical, Metabolic, and Neural Sciences, University of Modena and Reggio Emilia, 41125 Modena, Italy; francesco.ricchi@unimore.it (F.R.); stefania.caramaschi@unimore.it (S.C.); 2Pathology Unit, Department of Medical and Surgical Sciences for Children and Adults, University of Modena and Reggio Emilia, 41125 Modena, Italy; luca.fabbiani@unimore.it; 3Molecular Microbiology and Virology Unit, University Hospital Policlinico, 41124 Modena, Italy; sala.arianna@aou.mo.it; 4Department of Biomedical, Metabolic, and Neural Sciences, University of Modena and Reggio Emilia, 41125 Modena, Italy; laura.franceschini.bio@gmail.com; 5Department of Surgery, Dentistry, Morphological Sciences Related to Transplant, Oncology and Regenerative Medicine, University of Modena and Reggio Emilia, 41125 Modena, Italy; andrea.ardizzoni@unimore.it (A.A.); elisabetta.blasi@unimore.it (E.B.)

**Keywords:** dual infections, synthetic vaginal fluid, vaginal infections, A-431 cells, pathogens interactions

## Abstract

Little is known about the complex events driving host–pathogen and pathogen–pathogen interplay in polymicrobial infections. Using an in vitro model of a reconstituted vaginal epithelium (RVE) employing the A-431 cell line supplemented with synthetic vaginal fluid (SVF), we studied the consequences of single versus dual infections with *Candida albicans* and/or Herpes Simplex Virus-2 (HSV-2). Our data show (a) a relevant, SVF-enhanced expression of the differentiation marker cytokeratin 5/6 in the RVE; (b) the ability of *Candida albicans* to enhance HSV-2 in the dual infection model, with the virus titer almost doubling in the presence of SVF; (c) RVE damage (>20%), mostly attributable to *Candida albicans* and related to oxidative stress whether SVF is present; (d) the dysregulation of mucin-1, the production of which is enhanced (from 13 to 21 ng/mL) or impaired (from 21 to 10 ng/mL) in response to either SVF or infection, respectively; and (e) a partial-to-negligible cytokine response from the RVE, depending upon SVF presence. In conclusion, using an in vitro RVE model upgraded through the addition of synthetic vaginal fluid, we provide details on epithelial cell–pathogen–pathogen interaction, contributing to a better comprehension of the pathogenesis of polymicrobial infections at a mucosal level.

## 1. Introduction

Genital infections represent a significant public health concern worldwide. Among them, Herpes Simplex Virus-Type 2 (HSV-2) and *Candida albicans* are two of the most prevalent pathogens. HSV-2 primarily causes genital herpes, while *C. albicans* is the leading cause of vulvovaginal candidiasis [1,2]. Genital herpes is characterized by painful vesicular lesions in the genital region, although infections can be asymptomatic, leading to an underestimation of their prevalence. The virus establishes latency in the sacral ganglia and can be reactivated due to various triggers, including stress, immunosuppression, and co-infections [3]. HSV-2 infection is endemic globally, with estimates indicating that approximately 12% of the population aged 15–49 years is seropositive for HSV-2 [4]. This prevalence varies with demographic factors, including age, sex, and geographical location [5]. *C. albicans* is a commensal fungus found in various body sites, including the gastrointestinal tract, mouth, and vagina. Overgrowth of this fungus leads to candidiasis; substantial clinical manifestations depend on the site of infection and on the host’s condition [6]. Common presentations include vulvovaginal candidiasis, oropharyngeal candidiasis (thrush), and deep-seated/invasive candidiasis in immunocompromised individuals. Symptoms of vulvovaginal candidiasis include itching, burning, and a thick, white discharge, which may overlap, at least partly, with the initial symptoms of HSV-2 infection [7]. The incidence of vulvovaginal candidiasis is estimated to be around 75% in women during their lifetime, with recurrent infections affecting approximately 20–25% of these women [8].

Coinfections are an increasingly recognized phenomenon, particularly in patients with weakened immune systems, resulting in severe and difficult-to-treat diseases [9]. Some studies have shown that the presence of a primary infectious agent can predispose the host to a second infection, likely also implying pathogen-to-pathogen synergistic interplay [10,11]. Among various combinations of genital dual infections, *C. albicans* species and HSV-2 are noteworthy due to their overlapping risk factors and shared propensity to exploit host immunosuppression. Increasing data indicate that HSV-2 may enhance fungal colonization by affecting the integrity of the mucosal barrier and immune responses, as well as the host’s microbiome. As an example, HSV-2-induced epithelial disruption facilitates fungal adhesion and invasion, increasing susceptibility to secondary infections [12,13]. Moreover, cytokines produced during HSV-2 infection may also enhance *C. albicans* virulence, suggesting a complex interplay between the immune response elicited by HSV-2 and *C. albicans* pathogenic potential [14].

Although poorly investigated, genital co-infection with HSV-2 and *C. albicans* is emerging as an important area of interest. Co-infection, occurring especially among patients with repeated episodes of genital herpes, may lead to more severe symptoms and complications, necessitating attentive management strategies, simultaneously tailored towards both HSV-2 and *C. albicans* [15,16]. Antiviral therapy for HSV-2 and antifungal treatment for *C. albicans* should be considered concurrently, emphasizing the importance of integrated care and differential diagnoses. Additionally, the psychological impact of recurrent HSV-2 episodes can lead to increased stress and immunosuppression, further predisposing individuals to candidiasis [17]. Therefore, increased knowledge is needed on the interplay between these two pathogens to improve prompt and complete diagnosis and tailored treatment and, in turn, ameliorate patient management. Further research is imperative to elucidate the mechanisms behind their interactions, potentially leading to improved therapeutic strategies and patient outcomes.

Most pathogens enter the human body through mucosal barriers, which represent the first environment to be breached prior to establishing a successful infection [18]. Animal models have greatly improved our understanding of the initial steps of microbial pathogenesis; however, numerous limitations occur when assessing mucosal infections, given the complexity of the microenvironment and the multiplicity of cross-talks among different pathogens and hosts [19]. Here, we report an in vitro model of human epithelial cell line differentiated into a reconstituted multilayered epithelium, using which we evaluated susceptibility to single or dual infection by HSV-2 and *C. albicans*, virus vs. fungus growth, cell damage, secretion profile, and oxidative stress.

## 2. Materials and Methods

### 2.1. Epithelial Cells

All the experiments were carried out on the A-431 cell line, from a human epidermoid skin carcinoma (ATCC CLR-1555); such a cell line has been widely used to produce monolayers or multilayers mimicking the physiological epithelium [19,20]. The cells were cultured in DMEM (Sial S.r.l., Rome, Italy) supplemented with L-glutamine (2 mM) (Gibco, Thermo Fisher Scientific Italia, Segrate, Italy), 100 U/mL Penicillin–Streptomycin (Lonza Walkersville Inc., Walkersville, MD, USA), ciprofloxacin (2.5 mg/mL) (Gibco, Fisher Scientific Italia, Segrate, Italy), and heat-inactivated fetal bovine serum (Sial S.r.l., Rome, Italy) at 10% (growth medium, used at cell seeding) or 5% (maintenance medium, used after 5 days of cell growth, during the experimental procedures). The cell line was kept viable by subculturing twice a week and incubation at 37 °C with 5% CO_2_. The cells were used between the 30th and the 40th sub-culture passage.

### 2.2. Fungal Strain and Growth Conditions

The reference strain *C. albicans* SC5314 (ATCC MYA-2876) was employed. The fungal strain was stored as frozen stocks at −80 °C in Sabouraud Dextrose Broth (Condalab, Madrid, Spain) supplemented with 15% glycerol. After thawing, the fungal cells were grown in a liquid YPD medium (Scharlab S.L., Barcelona, Spain) and incubated at 37 °C under aerobic conditions for 24 h. Fungal cultures in the exponential growth phase were used in each experiment.

### 2.3. HSV-2 Strain

A clinical isolate of HSV-2, initially identified by a monoclonal antibody and laboratory-adapted through serial passages (>50) on Vero cells, was used. The viral suspensions used in the experiments were obtained upon the centrifugation of lysates of virus-infected Vero cells, which had been cultured for 2–3 days and showed a diffused cytopathic effect (more than 80% of monolayer destruction), as detailed elsewhere [21]. Prior to being used, the virus batches were titrated on A-431 cells, aliquoted, and kept frozen at −80 °C; all the experiments were carried out using the same batch.

### 2.4. Establishment of A-431 Monolayer Cultures and Reconstituted Epithelium (RVE)

Cells were organized as monolayers or as reconstituted vaginal epithelium multilayers, with cultures initially seeded at 4 × 10^5^ cells/mL (1 mL per well) in 24-well plates (Corning, New York, NY, USA). For the epithelial monolayers, A-431 cells were incubated overnight at 37 °C with 5% CO_2_, while, to obtain the RVE cultures, the A-431 cells were incubated for 5 days, with a fresh growth medium replacement at day 3.

### 2.5. Immunohistochemical (IHC) Staining for Cytokeratin 5/6 Detection

The monolayers and RVE cultures were processed for immunohistochemistry (IHC), with cells plated at 2 × 10^5^ cells/mL (400 μL per well) in chamber slides, following the same seeding and incubation protocol as used for the 24-well plates. After 1 or 5 days (for the epithelial monolayer or RVE, respectively), the slides were fixed in 95% ethylic alcohol for 10 min. IHC staining for cytokeratin 5/6 (CK 5/6) was performed using the anti-cytokeratin 5/6 mouse monoclonal primary antibody, clone D5/16B4, at an approximate concentration of 10.4 μg/mL (Roche Diagnostics-Ventana Medical Systems, Tucson, AZ, USA). Staining procedures were conducted using BenchMark IHC/ISH automated instruments (Ventana Medical Systems, Tucson, AZ, USA) according to standard antigen retrieval protocol. Antibody reactivity was visualized using a Ventana OptiView Universal DAB IHC Detection Kit and 3,3′-diaminobenzidine (DAB) chromogen (Roche Diagnostics—Ventana Medical Systems, Tucson, AZ, USA). Subsequently, the slides were counterstained with Hematoxylin II—a modified Mayer’s hematoxylin (Roche Diagnostics—Ventana Medical Systems, Tucson, AZ, USA)—to highlight cell morphology. Cells exhibiting membranous and/or cytoplasmic brown staining were considered positive for cytokeratin 5/6. Each well was scored as negative (<10% of positive cells), partially positive (10–80%), or diffuse expression (>80%). The intensity of staining was semi-quantitatively graded as weak (1), moderate (2), or strong (3).m

### 2.6. Infection Protocol

The infection protocol included the use of the monolayers and the RVE cultures, replaced with fresh maintenance medium before the infection; in selected wells, synthetic vaginal fluid (SVF) was also added (10% vol/vol), according to the protocol of Owen et al. and Del Gaudio et al. [22,23]. The SVF, intended to simulate human vaginal fluid, had the following formulation for 1 L of solution, given as a compound and its weight (g): calcium, 0.120; potassium, 0.978; sodium, 1.38; chloride, 2.13; albumin, 0.018; lactic acid, 2.00; acetic acid, 1.00; glycerol, 0.16; urea, 0.4; and glucose, 5.0. The final pH value was 4.2. Next, the cultures were infected with *C. albicans* (250 μL of fungal suspension/well), at a Multiplicity of Infection (MOI, fungal cells:epithelial cells) of 0.5:1; after a 3 h incubation at 37 °C, 250 μL of the HSV-2 suspension was added to a final MOI (virus: cells) of 0.1:1. The final working volume was 1 mL/well. Then, the cultures were incubated for a further 21 h at 37 °C prior to being assessed as detailed below.

### 2.7. Evaluation of Microorganisms-Induced Cell Damage by Quantification of LDH Release

Epithelial cell damage was quantified by analyzing the lactate dehydrogenase (LDH) release in the supernatant [24]; a commercially available kit (Roche, via Merk Life Science S.r.l., Milan, Italy) was employed, following the manufacturer’s instructions. The percentage of damage was calculated as follows:$$\%\;of\;cell\;damage=\frac{test\;sample-low\;control}{high\;control-low\;control}\times 100$$
where test sample is the O.D. value of the sample, low control is the mean value of uninfected cells, and high control is the mean value of uninfected cells lysed with 1% (vol/vol) Triton-X-100 (Fluka Chemicals, via Merk Life Science S.r.l., Milan, Italy) [25]. In selected experiments, the LDH assay was compared with the MTT assay (see Appendix A).

### 2.8. Pathogen Growth Quantification

*C. albicans* quantification on the epithelial cells was assessed using the CFU assay. At 24 h, cells were lysed with Triton X-100 (a final concentration of 0.1%), and serial dilutions from each well were performed and then seeded on Sabouraud agar plates, which were further incubated for 24 h at 37 °C. The resulting CFUs were counted. Each sample was assessed in triplicate.

For HSV-2 load quantification, a commercial Real Time-PCR was used (Elite InGEnius SP200, Elitech, Torino, Italy) after DNA extraction by means of a commercial kit (HSV2 ELITe MGB^®^ Kit, Elitech, Torino, Italy), according to the manufacturer’s instructions. The PRC protocol was as follows: decontamination at 50 °C/2 min, initial denaturation at 94 °C/2 min, and then at 94 °C/10 s, 60 °C/30 s, and 72 °C/20 s for 45 cycles.

In selected experiments, the results of the molecular assay were compared with those obtained by a plaque reduction assay, performed as previously described [21]. The rate of HSV-2-infected RVE cells was also checked with an indirect immunofluorescence assay using a monoclonal antibody against a HSV-2 capsid antigen (Merck Millipore, Milan, Italy), and a representative image is presented in the Appendix A.

### 2.9. Oxidative Stress Determination

The production of mitochondrial reactive oxygen species (mtROS) by epithelial cells, infected as detailed above, was measured at time 24 h. MitoSOX™ Red (2.5 μM/well) (Invitrogen™, Fisher Scientific Italia, Segrate, Italy) was added to each well immediately after infection. Then, the plates were evaluated using Fluoroskan (Thermo Scientific, Segrate, Italy) at 37 °C, for 24 h; the fluorescence emission was analyzed at an excitation/emission length of 544/590. The results at 24 h were depicted as a heatamp.

### 2.10. Quantification of IL-1α, IL-1β, IL-8 and Mucin-1 Production

The levels of IL-1α, IL-1β, IL-8, and mucin-1 were measured in 100 μL of A-431 cell-free supernatants at 24 h, using specific commercially available ELISA kits (PeproTech™, Fisher Scientific Italia, Segrate, Italy for IL-1α; Invitrogen™, Fisher Scientific Italia, Segrate, Italy, for IL-1β and IL-8; Elabscience Biotechnology, Huston, TX, USA for mucin-1) according to the manufacturer’s instructions.

### 2.11. Statistical Analysis

Unless differently indicated, each experiment was repeated 4–6 times, and the samples were assessed in triplicate. The Shapiro–Wilk test was used to analyze the data distribution within each experimental group. Subsequently, statistical analysis was performed using one-way ANOVA or the Kruskal–Wallis test, depending on the distribution of data; Tukey’s multiple comparisons were chosen as post hoc tests. For the heatmap obtained in the mtROS assay, the Area Under the Curve (AUC) was calculated to summarize the curve into a single value. Subsequently, statistical analysis was performed comparing the AUC values of each experimental group, using one-way ANOVA. All statistical analyses were carried out using GraphPad Prism 10 software. Values of * *p* < 0.05 and ** *p* < 0.01 were considered statistically significant.

## 3. Results

### 3.1. Cytokeratin-5/6 Expression in A-431 Epithelial Cells Maintained at Different Culture Conditions

Initially, we investigated the expression of cytokeratin 5/6, a well-established marker of epithelial cell differentiation. Thus, A-431 cells, cultured for 1 day (monolayer) or 5 days (RVE) with or without SVF, were processed and stained for IHC analysis. As detailed in Figure 1, the RVE showed a strong and diffused expression of this marker (intensity grade 3) in comparison to the cell monolayer (intensity grade 2); a further enhancement of positivity was observed if the RVE had been cultivated in the presence of SVF. Importantly, in this latter case, a more marked localization of the cytokeratin at the cell membrane level was evident.

### 3.2. Candida albicans and HSV-2 Load in A-431 RVE Infected with or Without SFV

The amount of fungal cells (expressed as CFUs) and virus production (expressed as DNA copies) was assessed in the RVE, infected with a single pathogen or both, in the presence and the absence of SVF. Figure 2A shows that *C. albicans* growth was significantly higher (>1 Log difference in CFU) in the samples cultured with SVF compared to the controls without SVF. Figure 2B shows the levels of HSV-2 DNA copies, determined using RT-PCR. The amount of viral DNA ranged between 3.9 × 10^5^ copies/mL and 7.3 × 10^6^ copies/mL, with no significant changes with respect to the presence of SVF. Yet, in the absence of SVF, the viral infectious titers slightly decreased in cells that had also been infected with *C. albicans* while, in contrast, in the presence of SVF, the HSV-2 DNA copies approximately doubled in dually infected cells with respect to A-431 cells infected with the virus alone. A similar trend of data was observed when the infectious titers were measured using the plaque assay (Figure 2C).

### 3.3. Damage of A-431 RVE upon Infection with One or Two Pathogens, in the Presence or Absence of SFV

The LDH assay was used to determine the levels of A-431 cell damage, induced by a single or a dual infection, in the presence or absence of SVF (Figure 3). HSV-2 alone did not affect epithelial cell viability (<2% of cytotoxicity), irrespective of the presence or absence of SVF. Conversely, *C. albicans* significantly damaged epithelial cells, with the cytotoxicity percentage rising from 16% without SVF to 24% in the presence of SVF. In the dually infected cultures, the LDH levels showed a further slight, insignificant increase. In selected experiments, the cell damage was also determined using the MTT assay, and the results were completely superimposable (see Appendix A).

### 3.4. Oxidative Stress in A-431 RVE Infected with Candida albicans and/or HSV-2, With or Without SFV

The A-431 RVE samples infected with *C. albicans* candida and/or HSV-2, in the presence or absence of SVF, were tested for oxidative stress and measured as mtROS production. The heatmap is shown in Figure 4. On the right line, the reference lane is depicted. The values indicated by the different colors represent the amounts of mtROS detected in each sample at 24 h. As shown in the figure, in all conditions, the ROS levels were higher in the samples with SVF compared to those without SVF. A clear additive effect was evident in the dually infected cells cultured with SVF, displaying levels that were approximately double those of the mono-infected samples. This was evident only when SVF was present.

### 3.5. Secretion Pattern of A-431 Epithelial Cells Exposed to Single or Dual Infection, With or Without SVF

Levels of IL-1α, IL-1β, IL-8, and mucin-1 were assessed in cell-free supernatants of A-431 cells infected with HSV-2 and/or *C. albicans* in the presence or absence of SVF. As shown in Figure 5, IL-1α and IL-1β were not detectable in the uninfected A-431 cells; induction occurred only by *C. albicans*, with no relevant differences when comparing mono and dual infection. As shown, in the presence of SVF, the production of IL-1α and IL-1β was greatly affected, with their levels being doubled (30 pg/mL vs. 60 pg/mL for IL-1 α and 6 pg/mL vs. 12 pg/mL for IL-1β). As for IL-8, this chemokine was basally produced by A-431 cells and significantly increased upon infection with *C. albicans*, but not with HSV-2. Unexpectedly, SVF caused a decrease in IL-8 levels (44 pg/mL vs. 28 pg/mL). As for mucin-1 production, enhanced levels were observed upon SVF addition (13 ng/mL vs. 22 ng/mL) in the uninfected cultures. In contrast, each infectious agent caused a decrease in mucin-1 production, which was significant for *C. albicans* (22 ng/mL vs. 11 ng/mL) but not for HSV-2 (22 ng/mL vs. 18 ng/mL). Moreover, the dual infection did not have any additive effect, since the mucin-1 levels remained almost unaffected (11 ng/mL with *C. albicans* alone; 13 ng/mL with both pathogens).

In the Appendix A, we provide a summary of RVE peculiarities with respect to SVF addition and infection with one or two pathogens.

## 4. Discussion

In the era of molecular diagnostics, increasingly sensitive techniques are showing how polymicrobial infections, at a mucosal level, are frequent and certainly underestimated events [9]. Scant research documents the frequency of double vaginal infections by HSV-2 and *C. albicans* [13,15], and even less is known about the pathogenic mechanisms involved and the possible synergistic action whereby one microorganism favors the other. We have previously demonstrated how *C. albicans* biofilm favors the persistence of HSV-1 and Coxsackie Virus B5 over time; importantly, the fungal pathogenic potential seems to increase further, since the presence of *C. albicans* biofilm protects viral infectious particles from the immune response, drugs, and disinfectants [26,27]. Similarly, the co-presence of HSV-1 or HHV-6 and *C. albicans* or *Cryptococcus neoformans* dysregulates monocyte-mediated immune functions [28,29].

Here, to study the interplay between HSV-2 and *C. albicans* at the mucosal level, we describe an in vitro epithelial model using the A-431 cell line, that depending on the culture conditions, forms a monolayer (1 day) or a differentiated stratified (5 days) epithelium [30]. The A-431 line exhibits a range of characteristics that are quintessentially typical of epithelial cells [31,32] including the high expression of keratins, structural proteins that not only support cellular architecture but also play a crucial role in the resilience and protective function of the epithelial barrier [33]. In the present study, to render our model as close as possible to the in vivo condition, the culture medium was treated with synthetic vaginal fluid (SVF), according to the previously established protocol [22,23]. Initially, we demonstrate that the 5-day RVE displays a strong and diffused expression of cytokeratin 5/6 with respect to the 1-day monolayers, and, interestingly, the addition of SVF further increases the expression of such a differentiation marker, especially in the RVE. Thus, hereafter, we focus on the RVE model as the condition providing the highest degree of epithelial cell differentiation.

The RVE appears to be a suitable in vitro model for studying single or dual infection by *C. albicans* or HSV-2. Indeed, we show that fungal growth occurs and significantly increases upon the addition of SVF. This result is not surprising since the acidic pH of the SVF and its high glucose concentration are parameters greatly favorable to *C. albicans* persistence and growth, both in vitro and in vivo. Interestingly, *C. albicans* CFUs are not affected upon HSV-2 co-infection, excluding any pro-*C. albicans* effect by the viral agent. Concerning viral quantification, SVF addition tends to reduce viral load. Possibly, changes in the extracellular environment induced by the fluid, including pH lowering, increased viscosity, and high levels of polysaccharides, may limit the initial epithelial cell–virus interaction, as shown in other infection models [34,35]. Moreover, in the co-infection, two opposite patterns of results have been observed with respect to SVF addition. In the absence of SVF, the viral load is reduced by the co-presence of *C. albicans*, while it is increased when the co-infection is performed with SVF. These data suggest that by utilizing the SVF content, fungal overgrowth may have led to fluid consumption and the restoration of conditions suitable for viral infection and replication. If this process has an in vivo counterpart, we can envision significant cooperation between fungal and viral agents that enhances their pathogenic potential at the mucosal level.

As established using the LDH release assay, epithelial cell damage is only attributable to *C. albicans*, as the contribution of HSV-2, alone or in co-infection, happens to be extremely limited regardless of the presence of SVF. This result may be due to the overgrowth of the fungus observed in cultures treated with SFV. The negligible contribution of the virus to epithelial cell damage may not be attributed to a low rate of viral infection since, as assessed using an immunofluorescence assay, more than 60% of the cells became infected (Appendix A). Likely, the lack of LDH release may be related to the short incubation time chosen in our protocol (24 h), as the inverted light microscope observation of the cell cultures reveals a limited cytopathic effect.

An imbalance in the redox state towards oxidant conditions is a key event during viral infections, also mediating tissue damage [36,37]. In particular, the involvement of ROS has been demonstrated in many viral infections, including HSV-1, where the dynamics of viral replication change, leading to increased viral loads and persistence within the host cell [38]. In the case of *C. albicans*, the presence of ROS correlates with reduced fungal viability and virulence, emphasizing their role as hosts’ essential defense mechanisms [39,40]. In our model, the epithelial cell infection by either *C. albicans* or HSV-2 alone causes a partial-to-negligible ROS response, while SVF alone increases ROS with respect to basal levels. Interestingly, in the presence of SVF, the two pathogens exert an additive effect with respect to ROS production, which is indeed more than doubled. Thus, we may conclude that A-431 epithelial cells, supported by SVF, better respond to the concomitant *C. albicans* and HSV-2 infection in terms of ROS production; however, this condition favors viral replication, as indicated by the viral loads that are augmented in the samples with double infection.

Besides oxidative stress, cytokines play a pivotal role in shaping the inflammatory response against infections. Among many, IL-1α and IL-1β are pro-inflammatory signals that crucially act in hosts’ defenses. Indeed, upon HSV-1 infection, epithelial cells release IL-1α that in turn induces necrosis and cell lysis. Meanwhile, IL-1α predominantly acts at the site of tissue damage, working as a further “alarmin” to locally signal injury [41,42]. In our model, the secretion of IL-1α does not change when the cells are infected with the virus alone, and this is in line with the limited cellular damage occurring in these same samples. Concerning IL-1β, it is known that such a cytokine is primarily secreted following activation of the inflammasome [43]. Interestingly, HSV-2 affects the activation of the NLRP3 inflammasome by two specific viral proteins [44] and, in turn, reduces IL-1β production. Our data show no changes in IL-1β production by HSV-2 in parallel with minor virus-mediated cell damage.

Increasingly, the literature has shown that IL-1α is released from epithelial cells during fungal invasions, acting as an early alarm signal to recruit immune cells [45,46]. Upon binding to specific pattern recognition receptors, *C. albicans* activates the NLRP3 inflammasome via NF-κB, in turn triggering IL-1β production [47]. In our model, both IL-1α and IL-1β are induced by *C. albicans* without any significant additive effect from the presence of HSV-2. Interestingly, upon the addition of SVF, IL-1α and IL-1β production is enhanced, indicating that this culture condition favors epithelial cell reactivity against *C. albicans*. Furthermore, IL-8, a well-known pro-inflammatory chemokine [48], shows no relevant variations in our model, irrespective of whether SVF is present or the epithelial cells have been infected by HSV-2 and/or *C. albicans*. Finally, mucin-1, released through the apical surface of epithelial cells, is crucial in protecting mucosal surfaces and provides barrier functions and immune regulation [49,50]. HSV-2 displays several mechanisms to evade mucin action, including downregulation of MUC1 gene expression [51]. Similarly, *C. albicans* can elude mucin-1 by secreting aspartyl proteases that promptly degrade mucins [52], allowing hyphal active penetration and tissue invasion through the mucosal barrier [53]. The results we obtained in our dual infection model add evidence for the ability of the two pathogens to modulate mucin-1 production. In particular, *C. albicans* and HSV-2, either alone or in co-infection, can reduce levels of mucin-1, the phenomenon being particularly evident in cultures with SVF.

Overall, using an in vitro RVE obtained from the A-431 cell line, we first show that *C. albicans* induces important cell damage, associated with a scant release of inflammatory cytokines and a relevant reduction in mucin, likely reflecting the complex double-face interplay between fungal and host cells occurring at the mucosal level. Second, superinfection with HSV-2 does not result in a modification of *C. albicans* behavior nor in a further increase in epithelial cell damage and response. Third, *C. albicans* favors viral replication, whose load in fact increases in cells already infected by *C. albicans* compared to uninfected cells, in the presence of the SVF. Bearing in mind the limitations of such an epithelial tumor cell-line model, we aim to implement it through (i) the addition of neutrophils, as the first line and most abundant immune cell recruited at the site of infection, and (ii) the use of synthetic scaffolds, as inert cell support, to produce three-dimensional culture structures, hopefully, better mimicking the in vivo architecture of the vaginal epithelium.

In conclusion, the A-431-based RVE represents a ductile and easy-to-perform model for in vitro studies on polymicrobial infections occurring at the mucosal level; the addition of SVF provides a first step ahead in mimicking the vaginal environment.

## Figures and Tables

**Figure 1 microorganisms-13-00905-f001:**
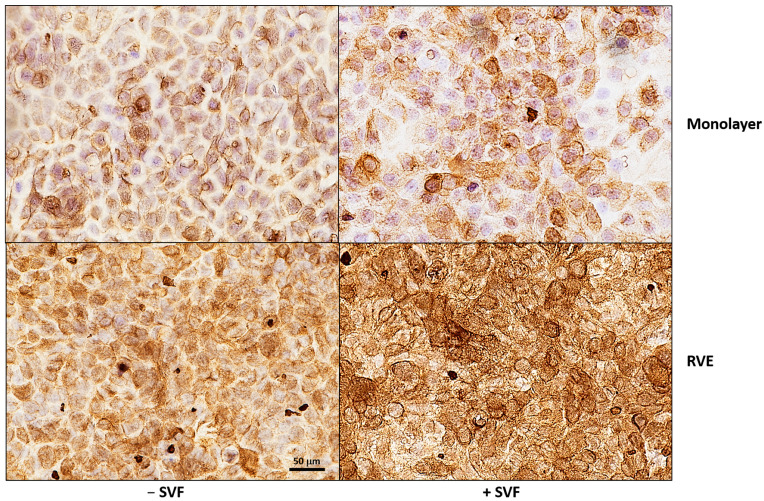
Cytokeratin 5/6 staining in A-431 cells, cultured for 1 or 5 days, in the absence or presence of SVF. Cells were immunohistochemically stained, with an anti-cytokeratin 5/6 mouse monoclonal primary antibody, using BenchMark IHC/ISH automated instruments according to standard antigen retrieval protocol. Representative images from two experiments are shown (magnification 40×). Immunohistochemical staining for CK5/6 shows diffused expression in the RVE, with stronger intensity observed in SVF.

**Figure 2 microorganisms-13-00905-f002:**
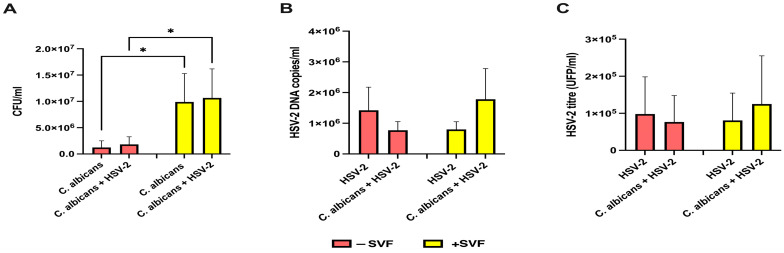
Quantification of fungal and viral loads in the RVE cultured with or without SVF. Five-day RVE cultures were infected with *C. albicans* (fungus:epithelial cell ratio = 0.5:1), and, after 3 h, HSV-2 (virus:cell ratio = 0.1:1) was added. Fungal and virus quantification was carried out after 24 h incubation using the CFU method on Sabouraud agar for *C. albicans* (**A**) and both RT-PCR (**B**) and the plaque reduction assay (**C**) for HSV-2. * *p* < 0.05.

**Figure 3 microorganisms-13-00905-f003:**
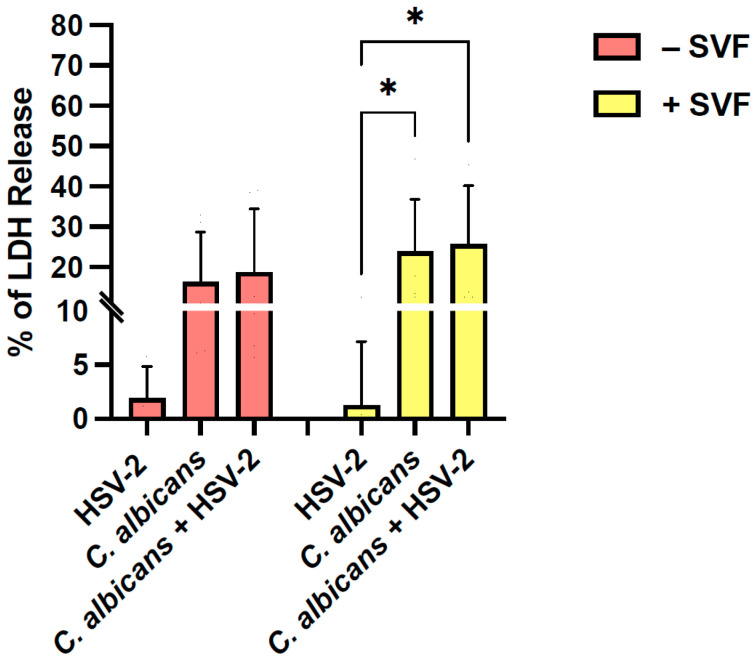
Levels of A-431 cell damage after single or dual infection in the presence or absence of SVF. Five-day RVE cultures were infected with *C. albicans* (fungus:epithelial cell ratio = 0.5:1), and, after 3 h, HSV-2 (virus:cell ratio = 0.1:1) was added. At 24 h, the LDH assay was performed to quantify the percentage of LDH release in the RVE exposed to one or two pathogens, in the presence or in the absence of SVF. * *p* < 0.05.

**Figure 4 microorganisms-13-00905-f004:**
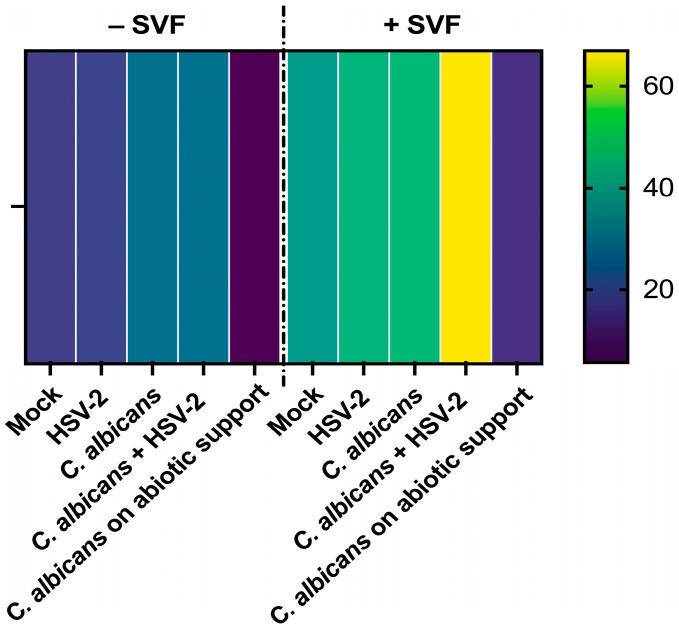
Mitochondrial ROS production by A-431 cells, exposed to single or dual infection in the presence or absence of SVF. Five-day RVE cultures were infected with *C. albicans* (fungus:epithelial cell ratio = 0.5:1), and, after 3 h, HSV-2 was added (virus:cell ratio = 0.1:1). Then, MitoSOX™ Red was added to each well and the plates were evaluated using Fluoroskan at 37 °C, for 24 h. A representative heatmap of the quantification of mtROS production at 24 h is shown. The mock line refers to A-431 cells alone; an inoculum of *C. albicans* without cells was included as a negative control (abiotic support). The values corresponding to the different colors represent the Area Under the Curve calculated to summarize the curve of the 24 h determination into a single value.

**Figure 5 microorganisms-13-00905-f005:**
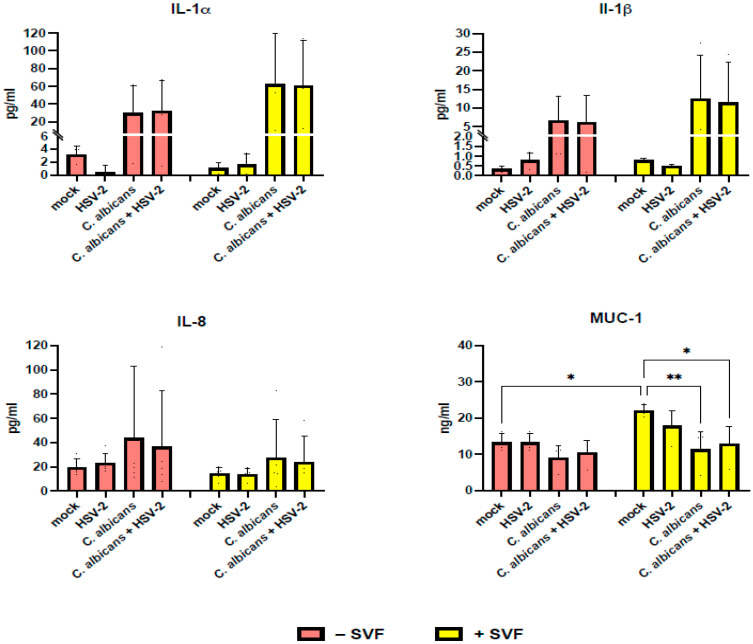
Secretory profile of the RVE infected with *C. albicans* and/or HSV-2, in the presence or absence of SVF. Five-day RVE cultures were infected with *C. albicans* (fungus:epithelial cell ratio = 0.5:1), and, after 3 h, HSV-2 was added (virus:cell ratio = 0.1:1). Following 24 h incubation, quantification of the secretion product was performed on the supernatants using commercial ELISA. * *p* < 0.05; ** *p* < 0.01.

## Data Availability

The original contributions presented in this study are included in the article/Appendix A. Further inquiries can be directed to the corresponding author.

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
