# Peer review of "Candida albicans as a Trailblazer for Herpes Simplex Virus-2 Infection Against an In Vitro Reconstituted Human Vaginal Epithelium"

_microorganisms, 2025, doi:10.3390/microorganisms13040905_

Round 1

Reviewer 1 Report

Comments and Suggestions for Authors

The study advances our understanding of polymicrobial infections by exploring the interactions between host-pathogen and pathogen-pathogen in an in vitro model of reconstituted vaginal epithelium. The authors demonstrated significant insights, such as Candida albicans facilitating Herpes Simplex Virus-Type 2 in the presence of synthetic vaginal fluid, and dysregulation of Mucin-1 production. Congratulations to the researchers for their dedication and for establishing this in vitro model, which provides a valuable tool for dissecting complex events underlying polymicrobial infections at the mucosal level.  The manuscript needs to be carefully revised before its publication.

  1. The abstract needs to be rewritten. The findings are described in a long sentence, which impair a clear understanding.
  2. Please avoid including opinion in the manuscript. The conclusion must be based on the findings.
  3. Please avoid using keywords used in the title.
  4. The abstract section provides a qualitative overview; however, the study includes quantitative analysis that offers a more comprehensive explanation of the findings. Please include a summary of them in the abstract.
  5. Sentences based on the literature require references. There are many sentences without citations. For example, lines 40, 40-42, 50-51, 51-53, 53-55, 55-57 and so on. I checked this in the first paragraph, but there are more in other parts of the current document. Please double check the entire manuscript.
  6. Line 99. Please specify passage number for cell lines used in the experiments.
  7. Line 104. Replace, by .
  8. Why were two different serum concentrations used? Why not only one?
  9. line 167. Please check spelling the formula.
  10. Line 193.Please clarify these cytokines and not others? Is there any rationale behind this selection?
  11. Figure 1. Scale bar must be added.
  12. How was the intensity grade estimated? What is this scale? Please clarify it. Can the authors used Image J or other tools to quantify and present this data in a graph with appropriate statistical analysis?
  13. The authors used the same colour for different conditions, which is quite confusing.
  14. Line 205. The values presented in the statistical section differ from those in the figure legends. They must be consistent.
  15. Figure 3. Although LDH is an indirect measurement of cell damage, the authors should label the graph as 'LDH Release.' In fact, they have measured LDH release in comparison to Triton-X.
  16. Please include units for heat maps (%)?
  17. Line 292. Scientific names must be italicised.
  18. The summary made can be presented as a figure in the discussion section or presented in the supplementary material. But is not relevant repeat the same in the results section. The compilation is more appropriate in the discussion section.
  19. The tables and figure legends must be enriched with more details of the methodlogy approaches.
  20. Line 385. If MTT was performed, please include as supplementary material. The same for line 396. Please avoid data not shown. If they are mentioned included them in the supplementary material.,
  21. References must be standardised. Sometimes, all words of the titles are capitalised and others not.
  22. I am curious about the limitations of the model used. They must be discussed. The research is fantastic but it is good to know the gaps between this model and in vivo studies.
  23. The statistical analysis seems not to be performed using the most appropriate test. If the comparison is with a group, another post-test hoc must be selected.

Author Response

We sincerely appreciate the constructive feedback and valuable insights provided by Reviewer 1. Below, we provide a detailed response to each of the reviewer’s points and outline the corresponding revisions made to the manuscript.
1.    The abstract needs to be rewritten. The findings are described in a long sentence, which impair a clear understanding. 

According to this comment, and in line with the request of Referee n. 2, the abstract has been extensively modified.
Little is known about the complex events driving host-pathogen and pathogen-pathogen interplay in polymicrobial infections. By an in vitro model of reconstituted vaginal epithelium (RVE) employing the A-431 cell line supplemented with a synthetic vaginal fluid (SVF), we studied the consequences of single versus dual infections with Candida albicans and/or Herpes Simplex Virus- 2 (HSV-2). Our data show: a) a relevant, SVF-enhanced expression of the differentiation marker Cytokeratin 5/6 in the RVE; b) the ability of Candida to advantage HSV-2 in the dual infection model, the virus title being almost double in the presence of SVF; c) the RVE damage (> 20 %), mostly attributable to Candida and related to oxidative stress whether SVF is present; d) the dysregulation of Mucin-1, whose production is enhanced (from 13 to 21 ng/ml) or impaired (from 21 to 10 ng/ml) in response to either SVF or infection, respectively; e) a partial to irrelevant cytokine response by RVE  occurs, depending upon SVF presence. In conclusion, by an in vitro RVE model upgraded by the addition of a synthetic vaginal fluid, we provide details on epithelial cell-pathogen-pathogen interaction, opening to a better comprehension of the pathogenesis of polymicrobial infections at mucosal level. Besides causing epithelial cell damage, Candida acts also as a facilitator of HSV-2. In our opinion, the model appears useful to further dissect the complex events underlying polymicrobial infections at mucosal level.

2.    Please avoid including opinion in the manuscript. The conclusion must be based on the findings.
We cut all the personal comments not based on the data obtained.
3.    Please avoid using keywords used in the title.
We eliminated the key words already present in the title
Keywords: dual infections; synthetic vaginal fluid; vaginal infections; A-431 cells; pathogens interactions
4.    The abstract section provides a qualitative overview; however, the study includes quantitative analysis that offers a more comprehensive explanation of the findings. Please include a summary of them in the abstract.
In the abstract revision we reported quantitative finding as highlighted in yellow at point 1) of this letter.
5.    Sentences based on the literature require references. There are many sentences without citations. For example, lines 40, 40-42, 50-51, 51-53, 53-55, 55-57 and so on. I checked this in the first paragraph, but there are more in other parts of the current document. Please double check the entire manuscript. 
We added 10 new references, mostly in the Introduction section, but also in M&M and Discussion.
6.    Line 99. Please specify passage number for cell lines used in the experiments.
We added this sentence: The cells were used between the 30th and the 40th sub-culture passage.
7.    Line 104. Replace, by .
Sorry, we did not understand this comment. At line 104 there is just a list of additives for the growth medium.
8.    Why were two different serum concentrations used? Why not only one?
We specified in the test that 10% is for the growth medium, used when we seed cells, while 5% is for the maintenance medium, used after infection when cells already reached confluence.
9.    line 167. Please check spelling the formula.
We corrected the misspelling and clarified the meaning of the terms present in the formula
10.    Line 193.Please clarify these cytokines and not others? Is there any rationale behind this selection? 
The choice derives from previous works, where the A-431 cells showed some degree of fluctuation on these cytokines.
11.    Figure 1. Scale bar must be added.
We added the scale bar in Figure 1.
12.    How was the intensity grade estimated? What is this scale? Please clarify it. Can the authors used Image J or other tools to quantify and present this data in a graph with appropriate statistical analysis?
The stained slides were evaluated under the light microscope by trained researcher. The intensity of staining was semi-quantitatively graded as weak (1), moderate (2), or strong (3) for each well. We appreciate the reviewer’s suggestion to quantify this expression; however, we currently do not have access to the necessary computational tools to perform this type of quantitative analysis. The semiquantitative approach we employed is widely used in the anatomopathological field to convert subject perception of IHC-marker expression into semiquantitative data. We did not consider performing a statistical analysis on these data, as they represent an observational and preliminary assessment rather than the central focus of the research. Nonetheless, we recognize the value of such an analysis and will consider its inclusion in future studies.
13.    The authors used the same color for different conditions, which is quite confusing.
We acknowledge that the use of the same color for different conditions may lead to potential confusion; however, the selection of brown was not arbitrary. The reactivity of the anti-Cytokeratin 5/6 mouse monoclonal primary antibody was visualized using the Ventana OptiView Universal DAB IHC Detection Kit, coupled with the 3,3′-diaminobenzidine (DAB) chromogen. The peroxidase enzyme catalyzes the conversion of the DAB substrate into a brownish precipitate, which is deposited at the site of the reaction, thereby providing a visual representation of the location where the primary antibody binds to its antigen target. The insolubility and stability of the brown-colored DAB oxidation product allow for the subsequent mounting and archiving of the preparations according to standard histological techniques.
14.    Line 205. The values presented in the statistical section differ from those in the figure legends. They must be consistent.
We apologize for the typing error, that has now been corrected.
15.    Figure 3. Although LDH is an indirect measurement of cell damage, the authors should label the graph as 'LDH Release.' In fact, they have measured LDH release in comparison to Triton-X.
We changed the term cytotoxicity with LDH release both in the text and in the figure.
16.    Please include units for heat maps (%)?
The colors in the scale corresponds to different values of the area under the curve calculated to summarize the curve of the 24h determination into a single value.
17.    Line 292. Scientific names must be italicised.
Sorry for the mistyping error, we corrected it.
18.    The summary made can be presented as a figure in the discussion section or presented in the supplementary material. But is not relevant repeat the same in the results section. The compilation is more appropriate in the discussion section.
We agree that the table and its comment make the text excessively long and redundant so we have moved the table to the supplementary material section
19.    The tables and figure legends must be enriched with more details of the methodlogy approaches.
We added technical and/or methodologic information
20.    Line 385. If MTT was performed, please include as supplementary material. The same for line 396. Please avoid data not shown. If they are mentioned included them in the supplementary material.,
We added pictures of the immunofluorescence assays in the supplementary material section as well the results of a comparison between MTT and LDH.
21.    References must be standardised. Sometimes, all words of the titles are capitalised and others not.
We acknowledge the inconsistency in capitalization and appreciate the reviewer’s observation. However, we would like to clarify that the references have been reported exactly as they appear in PubMed or as provided by the respective journals for citing the articles. Nevertheless, should it be deemed necessary, we will standardize the references in accordance with the required citation style in the final version of the manuscript.
22.    I am curious about the limitations of the model used. They must be discussed. The research is fantastic but it is good to know the gaps between this model and in vivo studies.
Thanks for the comment. We added the following sentence indicating how we would proceed to improve the model or, in other words, to overcome some of the current limitations.
Bearing in mind that this is a model based on a tumor cell line with all the limitations that this implies, in attempt to further implement it, we aim to i) add neutrophils, as the first line and most abundant immune cell recruited at the site of infection, and ii) use synthetic scaffolds, as inert cell support, to produce a three-dimensional structure of the cultures, hopefully better mimicking the in vivo architecture of vaginal epithelium.
23.    The statistical analysis seems not to be performed using the most appropriate test. If the comparison is with a group, another post-test hoc must be selected. 
We decided to compare the mean of each column (representing an experimental group) with the mean of every other column. Next, we applied a correction for multiple comparisons using statistical hypothesis testing. Tukey's test is used when the distribution is normal and the initial test is performed with one-way ANOVA. Conversely, Dunn's test is applied when the distribution is not normal and the first test is conducted using the Kruskal-Wallis test.

Reviewer 2 Report

Comments and Suggestions for Authors

The manuscript refers to the role of Candida albicans on HSV-2 infection of the human vaginal epithelium. Even though the topic is relevant from the clinical point of view, the experimental design is not well established. If Candida enhances HSV-2 infection, then the logical cytokine to monitor is IFNalpha. Also, Candida forms a film on the top of the cells in culture, so the decrease in MUC-1 observed is expected. In addition, it would be interesting to do a short priming of the cells in culture with LPS before infection since epithelial cell antigen expression may be modified and the infection rate would also de modified.  The authors may use candidin instead see  DOI: 10.3390/molecules20022272. The results of cytokine determination are different from previously reported see DOI: 10.3390/molecules27030782

What is the fundamental role of mitochondrial metabolism in cells in hypoxic conditions? If the cells were incubated with low glucose, what would be the effect of mitochondrial metabolism? 

The results do not support the long and repetitive discussion. Please modify it.

Author Response

We thank the Reviewer 2 for the insightful comment and valuable observations regarding our manuscript.
The manuscript refers to the role of Candida albicans on HSV-2 infection of the human vaginal epithelium. Even though the topic is relevant from the clinical point of view, the experimental design is not well established. 
We employed a model widely used and extensively described in the literature (A-431 cell line). We added a synthetic vaginal fluid (according to previous studies) with the aim of rendering the in vitro model at least one step closer to the in vivo vaginal environment. We are conscious of the limitations of the system. Yet, we believe that it is a precious in vitro tool for further studies on mucosal infections. Nevertheless, as described in a sentence, we aim to further upgrade the model.  “ Bearing in mind that this is a model based on a tumor cell line with all the limitations that this implies, in attempt to further implement it, we aim to i) add neutrophils, as the first line and most abundant immune cell recruited at the site of infection, and ii) use synthetic scaffolds, as inert cell support, to produce a three-dimensional structure of the cultures, hopefully better mimicking the in vivo architecture of vaginal epithelium.”
If Candida enhances HSV-2 infection, then the logical cytokine to monitor is IFNalpha. 
Thanks for the comment. Unfortunately, despite the many assays performed, only a few cytokine fluctuations were observed, as detailed in the paper (see figure 5). However, we tried to address this specific issue and carried out ELISAs on the supernatants of previous experiments using a commercial kit (Human IFN-a2 DuoSet ELISA, R&D Systems) and we report the results here. Unfortunately, all the experimental samples resulted lower than the minimum value of the standard curve. However, interestingly the values from the cultures treated with SVF were slightly higher, confirming the protective role of this fluid. Since we did not obtain results useful to enrich the discussion, we did not add them in the revised paper.

Also, Candida forms a film on the top of the cells in culture, so the decrease in MUC-1 observed is expected.
We shared the idea of the Referee. Indeed, we have attempted to measure biofilm formation in our model, but we have failed to obtain consistent data, so far; in any case, to our knowledge, we provide the first evidence on the reduction in mucin-1 production by epithelial cells upon Candida infection.
 In addition, it would be interesting to do a short priming of the cells in culture with LPS before infection since epithelial cell antigen expression may be modified and the infection rate would also be modified.  
Thanks to the Referee for the suggestion. Certainly, a lot must be done yet. By the present paper, we aimed to characterize the model, using basic protocols and to make it as similar to the physiological epithelium as possible, mimicking natural conditions of infection. If accepted by these Referees, the model will be extensively employed for a variety of different studies on microbial pathogenesis at mucosal level. A sentence has been added to this matter.
The authors may use candidin instead see  DOI: 10.3390/molecules20022272. The results of cytokine determination are different from previously reported see DOI: 10.3390/molecules27030782
We appreciate the suggestions; there is no doubt that by changing, even a little bit, the experimental conditions different scenarios may occur. Moreover, as said above, we tried to simulate natural occurring infection
What is the fundamental role of mitochondrial metabolism in cells in hypoxic conditions? If the cells were incubated with low glucose, what would be the effect of mitochondrial metabolism? 
We thank for the idea of focusing on these aspects, which were not our initial aim to set up and characterize this model; we hope we may soon have the opportunity to approach this kind of analysis.
The results do not support the long and repetitive discussion. Please modify it.
The discussion has been modified and shortened.

Reviewer 3 Report

Comments and Suggestions for Authors

In the manuscript, the authors propose a cell line model for investigating viral-fungal co-infections. While the results presented may not appear particularly striking, studying mixed infections poses significant challenges, especially in understanding also the host cell response. The authors have summarized their findings and, based on these observations, suggested potential directions for future research.

As a major comment, I recommend introducing into the manuscript additional data, which were mentioned in the discussion as data not shown and concern additional important observations from the MTT tests and virus propagation analysis.

Furthermore, please explain why these specific MOIs were chosen or were other ratios tested?

The composition of SVF should be fully explained in the materials and methods section of the manuscript. 

More details in the methodology should be included in general about applied protocols and conditions.

The genus name Candida used throughout the text to describe the species studied should be replaced by C. albicans.

The notation of units should be reviewed and unified in the materials and methods section.

Author Response

We sincerely appreciate the constructive feedback and valuable insights provided by Reviewer 3. Below, we provide a detailed response to each of the reviewer’s points and outline the corresponding revisions made to the manuscript.

In the manuscript, the authors propose a cell line model for investigating viral-fungal co-infections. While the results presented may not appear particularly striking, studying mixed infections poses significant challenges, especially in understanding also the host cell response. The authors have summarized their findings and, based on these observations, suggested potential directions for future research.

As a major comment, I recommend introducing into the manuscript additional data, which were mentioned in the discussion as data not shown and concern additional important observations from the MTT tests and virus propagation analysis.
We agree that we reported many data as data not shown. We added in the supplementary section the MTT data and the results of the immunofluorescence assay documenting a high rate of HSV-2 positive cells.

Furthermore, please explain why these specific MOIs were chosen or were other ratios tested?
 We tried different MOIs, for both fungal and viral infections and we choose those that avoided a rapid and complete destruction of the RVE.

The composition of SVF should be fully explained in the materials and methods section of the manuscript. 
We added the complete formulation

More details in the methodology should be included in general about applied protocols and conditions.
We added more details, enlightened in yellow in the M&M section

The genus name Candida used throughout the text to describe the species studied should be replaced by C. albicans.
 We changed it all over the text

The notation of units should be reviewed and unified in the materials and methods section.
We apologize for the lack of uniformity. We corrected all the notations.

Reviewer 4 Report

Comments and Suggestions for Authors

Comments

Ricchi F. et al, describes a novel model to study the co-infection of Candida albicans and HSV. The work is interesting and up to date, since both infections are common and the combination may have a clinical significance to patients. Overall, the work was well designed, results are presented in a good way, and conclusions supported by the results. After some corrections, the reviewer considers the work suitable for publication.

Major comment:

  1. Some information on the materials and methods is missing.
  2. Did you characterize the model prior to infections? Considering, for instance, protein expression.

Minor comments:

Line 2 – “Against an in vitro” or in an in vitro?

Line 25 – 33 – very long paragraph… Try to use (a), (1) or i).

Line 34 – Rephrase by changing the expression “In our opinion”

Line 105 – Why the different FBS %?

Line 104 – Can the ciprofloxacin on the medium have an impact on the assays with the virus or fungus?

Line 119 – The reference of the strain is needed

Line 136 – Which alcohol?

Line 137 – concentration

Line 217 – Avoid the use of words such as “massive”

Line 222 – Is it possible to quantify the expression?

Line 234 – The information on the RT-PCR is missing in the materials and methods

Line 242 – Use a, b, and C instead of left panel, central, right.

Line 256 – Is the cytotoxicity between C albicans and C albicans + HSV-2 statistical different?

Line 272 – very bad quality…

Line 321 – This table was introduced as a figure. Add the table1

Author Response

Ricchi F. et al, describes a novel model to study the co-infection of Candida albicans and HSV. The work is interesting and up to date, since both infections are common and the combination may have a clinical significance to patients. Overall, the work was well designed, results are presented in a good way, and conclusions supported by the results. After some corrections, the reviewer considers the work suitable for publication.

Major comment:

Some information on the materials and methods is missing.
We added more details, enlightened in yellow in the M&M section

Did you characterize the model prior to infections? Considering, for instance, protein expression.
 We employed a model widely used and extensively described in the literature (A-431 cell line).
Minor comments:

Line 2 – “Against an in vitro” or in an in vitro?
We used the term “against” to give the idea of a sort of war contrast, an invasion and damage by two pathogens

Line 25 – 33 – very long paragraph… Try to use (a), (1) or i).
 We modified the abstract using a more schematic writing.

Line 34 – Rephrase by changing the expression “In our opinion”
 We changed the text according to your suggestion
Line 105 – Why the different FBS %?
 We specified in the test that 10% is for the growth medium, used when we seed cells, while 5% is for the maintenance medium, used after infection when cells already have reached confluence.
Line 104 – Can the ciprofloxacin on the medium have an impact on the assays with the virus or fungus?
No, ciprofloxacin has no activity on fungi and viruses. We have years of experience in using this antibiotic in cell culture media to prevent bacterial contamination without any influence on cell metabolism and viral or fungal growth.

Line 119 – The reference of the strain is needed.
 We isolated this viral strain was from a case of male genital ulcer more than 30 years ago and was then frequently passaged on cell cultures. We recently published a paper on the antiherpetic activity of a peptide using this strain. Sala A, Ricchi F, Giovati L, Conti S, Ciociola T, Cermelli C. Anti-Herpetic Activity of Killer Peptide (KP): An In Vitro Study. Int J Mol Sci. 2024 Oct 1;25(19):10602. doi: 10.3390/ijms251910602.

Line 136 – Which alcohol?
Ethylic alcohol. We have now specified in the paper.

Line 137 – concentration
Specific antibody concentration is approximately 10.4 μg/ml, as detailed in the Manufacturer’s instructions. We added it in the manuscript.

Line 217 – Avoid the use of words such as “massive”
 We changed this term with “strong”. 

Line 222 – Is it possible to quantify the expression?
We acknowledge the Reviewer's suggestion to quantify this expression; however, we currently lack access to the necessary tools to conduct a quantitative analysis. The expression of CK 5/6 was evaluated by trained researcher with a semi-quantitative method, as widely employed in histopathological field. Each well was positive for the marker and the staining intensity was semi-quantitatively graded as weak (1), moderate (2), or strong (3). As we can see in Figure 1, IHC staining revealed a strong diffused expression in RVE cultivated with SVF. We added this sentence in caption: “Immunohistochemical staining for CK5/6 show a diffused expression in RVE, with a stronger intensity observed in SVF.”.

Line 234 – The information on the RT-PCR is missing in the materials and methods
  We added the details of the PCR protocol.

Line 242 – Use a, b, and C instead of left panel, central, right.
 We accordingly changed the figure.

Line 256 – Is the cytotoxicity between C albicans and C albicans + HSV-2 statistical different?
No, they are not. The only significant differences are between cells infected with virus alone and both cells infected with C. albicans and cells infected with both pathogens. No differences between the two groups with Candida.

Line 272 – very bad quality…
We agree and we up-loaded a better image.

Line 321 – This table was introduced as a figure. Add the table1
We had to convert the table in a picture because there are technical troubles with the file template and the table results completely screwed up. Anyhow, accordingly to another Referee who considers this table redundant, we put it in the Supplementary materials

Round 2

Reviewer 2 Report

Comments and Suggestions for Authors

The authors slightly improved the manuscript. The authors did not answer many queries and the discussion was not modified accordingly. 

Author Response

(The authors gave the same response as above.)

Reviewer 4 Report

Comments and Suggestions for Authors

The authors replied successfully to all my comments.

Author Response

(The authors gave the same response as above.)
